

# The Arctic sea ice cover of 2016: A year of record low highs and higher than expected lows

Alek A. Petty[1,2], Julienne. C. Stroeve[3,4], Paul R. Holland[5], Linette N. Boisvert[1,2], Angela C. Bliss[1,2], Noriaki Kimura[6], Walter N. Meier[4]

5   [1]Earth System Science Interdisciplinary Center, University of Maryland, College Park, MD, USA

[2]Cryospheric Sciences Laboratory, NASA Goddard Space Flight Center, Greenbelt, MD, USA

[3]Center for Polar Observation & Modelling, University College London, London, UK

[4]National Snow and Ice Data Center, Cooperative Institute for Research in Environmental Sciences, University of Colorado, Boulder, Colorado, USA

10   [5]British Antarctic Survey, Cambridge, UK

[6]Atmosphere and Ocean Research Institute, The University of Tokyo, Tokyo, Japan

*Correspondence to*: Alek A. Petty (alek.a.petty@nasa.gov)

**Abstract.**

2016 was an interesting year in the Arctic, with record low sea ice at the start of the year, but a summer (September) Arctic
sea ice extent that was higher than expected by most seasonal forecasts. Here we explore the 2016 Arctic sea ice state in terms of its monthly sea ice cover, placing this in context of the sea ice conditions observed since 2000. We demonstrate the sensitivity of monthly Arctic sea ice extent and area estimates, in terms of their magnitude and annual rankings, to the ice concentration input data (using two widely used datasets) and to the methodology used to convert concentration to extent (daily or monthly extent calculations). We use estimates of sea ice area to analyse the relative 'compactness' of the Arctic sea
ice cover, highlighting anomalously low compactness in the summer of 2016 which contributed to the higher than expected September ice extent. Two cyclones that entered the Arctic Ocean during August appear to have driven this low concentration/compactness ice cover, but were not sufficient to cause more widespread melt out and a new record low September ice extent. We use concentration budgets to explore the regions and processes (thermodynamics/dynamics) contributing to the monthly 2016 extent/area estimates highlighting, amongst other things, rapid ice intensification across the
central eastern Arctic through September. Two different products show significant early melt onset across the Arctic Ocean in 2016, including record early melt onset in the North Atlantic sector of the Arctic. Our results also show record late 2016 freeze up in the Central Arctic, North Atlantic. and the Alaskan Arctic sector in particular, associated with strong sea surface temperature anomalies that appeared shortly after the 2016 minimum (October onwards). We explore the implications of this low summer ice compactness for seasonal forecasting, suggesting that sea ice area could be a more reliable metric to forecast
in this more seasonal, 'New Arctic', sea ice regime.



## 1 Introduction

A dramatic indicator of global climate change is the accelerated loss of Arctic sea ice (Stroeve et al., 2012; Serreze and Stroeve, 2015; Notz and Stroeve, 2016). Over the last several decades, Arctic sea ice extent (SIE) has declined across all seasons, with the strongest decline observed in September; the end of the summer melt season (e.g. Serreze et al., 2007).

Indeed the 10 lowest months of September Arctic SIE have all occurred within the last 10 years. Global Climate Models (GCMs) and observations suggest the Arctic will become ice free in summer sometime during the middle of the century (e.g. Stroeve et al., 2012, Notz and Stroeve, 2016, Jahn et al., 2016).

This ice loss has profound consequences for the Earth system, including impacts on Arctic ecosystems (e.g. Post et al., 2013; Meier et al., 2014); potential changes to mid-latitude weather (e.g. Cohen et al., 2014; Screen et al., 2014; Francis et al.,

2015), and human activities in the Arctic. The human impacts in particular have prompted an increased need to improve Arctic sea ice forecasts on seasonal timescales (e.g. Eicken, 2013). In response to this increased interest in seasonal forecasting, the Study of Environmental Arctic Change (SEARCH) has led a grass-roots effort, since 2008, to collect and synthesize forecasts of pan-Arctic September SIE from the research community, resulting in an annual Sea Ice Outlook (SIO) report, compiled in recent years by the Sea Ice Prediction Network (SIPN). The outcomes of SIPN and the activities of

the wider Arctic sea ice community are of considerable interest to the media and general public, especially considering the role of Arctic sea ice as an indicator of global climate change. As such, providing an accurate assessment of the Arctic sea ice state, and better communicating sea ice variability/uncertainty is paramount.

Record high air temperatures and low sea ice were observed in the Arctic winter/spring of 2016, including low/record low SIE from January to June, high sea ice and ocean surface temperatures, and a thinner ice pack than recent winters (Boisvert

et al., 2016, Cullather et al., 2016; Overland and Wang, 2016; Petty et al., 2017; Ricker 2017). This led to heightened speculation regarding a potential new record low September Arctic SIE. In fact, a new record low September Arctic SIE was not suggested by the SIO in 2016, despite this strong winter/spring preconditioning. The median July SIO forecast for the 2016 September extent was 4.30 million $km^2$, higher than the record low September Arctic SIE of 3.63 million $km^2$ which was set in 2012. The median July SIO forecast ended up being 0.42 million $km^2$ below the 'observed' SIE, which was

reported by the National Snow and Ice Data Center (NSIDC) as 4.72 million $km^2$. The potential importance of wintertime sea ice preconditioning for summer sea ice is clearly still very uncertain. A similar discussion is emerging this year, 2017, as Arctic sea ice tracks close to what was observed in 2012 and 2016 (http://nsidc.org/arcticseaicenews/2017/07/arctic-ice-extent-near-levels-recorded-in-2012/).

The summer of 2016 also featured two storms that entered the Arctic, which were implicated in the anomalous behavior

observed in summer, making the forecasts of September SIE challenging (e.g. Petty et al., 2017). Historically, summers dominated by low sea level pressure anomalies and increased cyclonic activity within the central Arctic tend to result in less sea ice loss due to ice divergence and cooler temperatures (Screen et al., 2011), while summers with high sea level pressure





anomalies tend to result in clear skies and warmer air temperatures that enhance ice loss (e.g. Serreze et al., 2016). However, there is some suggestion that as the ice cover thins, the response to, or the importance of, summer weather patterns will change (e.g. Holland and Stroeve, 2011). In addition, the timing of when cyclonic activity occurs may also play a role in how the sea ice cover responds (Serreze et al., 2016). This was discussed briefly in the 2016 SIO post-season report

(https://www.arcus.org/sipn/sea-ice-outlook/2016/post-season), however a more detailed discussion of the 2016 summer storms is needed.

While considerable recent research efforts have focussed on understanding and predicting these rapid summer Arctic sea ice declines, the rapid rate of Arctic warming (commonly referred to as Arctic amplification) is, in fact, stronger in autumn, winter and spring (e.g. Pithan and Mauritsen, 2014, Cohen et al., 2014). The winter/spring sea ice declines in 2016 suggest

this warming may be having a more significant impact on sea ice than in previous years, with the autumn of 2016 featuring anomalously warm SSTs across the Arctic, which likely delayed ice freeze-up and contributed to low SIE in these autumn/winter months. The anomalous behaviour observed throughout 2016 motivates a more detailed analysis of the entire 2016 Arctic sea ice state, especially if the behaviour observed in 2016 becomes commonplace. We also seek to demonstrate when and why monthly record low Arctic sea ice states were observed across 2016 as this formed a significant part of the

discussion surrounding Arctic sea ice throughout the year. We focus on sea ice cover, not ice thickness/volume, due to the consistent long-term record available, and its interest to Arctic stakeholders and the sea ice prediction community. The paper is organized as follows: Section 2 presents the datasets used in this study; Section 3 discusses the methods employed to investigate the ice concentration budgets; Section 4 presents results and discussion from our various analyses; and concludary remarks are given in Section 5.

## 20 **2 Data**

### *2.1. Sea ice*

We utilize sea ice concentration (SIC) data derived from satellite passive microwave brightness temperature ($T_b$) observations. The $T_b$ observations are obtained from the Defence Meteorological Satellite Program (DMSP) , including the Nimbus-7 Scanning Multichannel Microwave Radiometer (SMMR, 1978-1987), the DMSP F8, F11 and F13 Special Sensor

Microwave/Imagers (SSM/Is, 1987-2008), the F17 Special Sensor Microwave Imager/Sounder (SSMIS, 2009 to May 2016) and the DMSP F18 SSMIS from April 2016 onwards. Significant uncertainties exist in the processing of passive microwave $T_b$ for estimating SIC, including challenges associated with low winter open water fractions (e.g., Kwok, 2002), and the interpretation of surface melt signatures in the $T_b$ data (several products and their differences are discussed by Ivanova et al., 2015). We thus choose to use both the NASA Team (Cavalieri et al., 1996, updated 2017) and Bootstrap (Comiso, 2000,

updated 2015) SIC datasets, which use different methods for converting $T_b$ to SIC. Note that for the 2016 we use the daily Bootstrap SIC data provided courtesy of J. Comiso. All SIC data are provided on a 25 km x 25 km polar stereographic grid.





Due to differences in satellite orbit and sensor characteristics, the $T_b$/SIC data feature a time-varying pole hole depending on the passive microwave sensor used, which broadly translates to a pole hole north of 84.5 $^o$N (1979–June 1987), 87.2 $^o$N (July 1987–December 2007), 89.2 $^o$N (2008 onwards). Differences between the NASA Team and Bootstrap algorithms have been well-explored (e.g., Comiso et al., 1997; Meier, 2005; Ivanova et al., 2015; Comiso et al., 2017). In general, Bootstrap is less

sensitive to summer surface melt because of the passive microwave channel combination it uses and because it employs daily-varying tie points (coefficients for 100% water and 100% ice). These differences can be significant in terms of absolute concentration and extent; however, trends and anomalies generally have much smaller differences (Comiso et al., 2017).

While the satellite passive microwave record extends back to late 1978, we focus mainly on data from 2000 onwards, to

explore recent changes in the context of the 'New Arctic' – the period broadly covering the recent period of lower Arctic sea ice (e.g. Serreze and Stroeve, 2015). We choose to primarily focus our analysis on the Bootstrap data, as it is less affected by summer melt, a time of particular focus. However, as noted above, trends and anomalies are more similar between NASA Team and Bootstrap and this selection is not thought to substantially change our analysis and conclusions. The raw monthly 2016 Bootstrap SIC maps are shown in Figure 1, with anomalies relative to the 2000-2015 mean monthly SIC shown in

Figure 2. Anomaly SIC maps using the NASA Team data are shown in Figure S1.

Monthly indices of Arctic sea ice extent (SIE) and sea ice area (SIA) are produced from the NASA Team SIC data and disseminated to the public by the National Snow and Ice Data Center (NSIDC) as the Sea Ice Index (version 2.1, Fetterer et al., 2016). As we wish to explore the differences in SIE from the two algorithms, we choose to calculate SIE from the raw SIC data, following the methodology of the NSIDC Sea Ice Index but applied to both NASA Team and Bootstrap SIC data.

Briefly, a monthly mean gridded SIC field is generated and a monthly ice flag dataset is used to discard grid cells that are thought to be incorrectly characterized as ice. For SIE, all grid cells with a SIC greater than 0.15 are set to 1, multiplied by the grid-cell area and summed together. All data within the variable pole hole are assumed to be ice covered (SIC=1) and are thus included fully in the SIE calculation. A comparison of our NASA Team SIE data with the NSIDC Sea Ice Index show small (~0.01-0.05) differences (the 2016 values are given in Table 1), which is thought to be due primarily to our use of the

daily SIC data, which are then averaged monthly (to be consistent with the 2016 Bootstrap data that is only available daily), instead of the monthly SIC data for years prior to 2016 (as the NSIDC does). This does have a small impact on our SIE rankings (as discussed later).

For SIA, the NSIDC Sea Ice Index approach is to not 'fill' the pole hole when calculating SIA, meaning the time series is significantly impacted by the changing size of the pole hole, especially for earlier years in the satellite record. We instead

apply a mean SIC calculated in a 0.5$^o$ halo around the variable pole hole to all grid cells within the pole hole, to crudely limit the bias introduced by the time varying pole hole size. All grid cells with a SIC greater than 0.15 are multiplied by the grid-cell area (the SIC is kept variable and not set to 1) and summed together. Note that a similar approach was used in the SIO





2016 post-season report. This approach of 'filling the pole hole' for longer-term analyses is discussed by, for example, Olason and Notz, (2014, Appendix A), where they show that this matters more for specific algorithms, e.g. the NASA Team algorithm, which has lower SICs in the Central Arctic, but is less important for the higher SIC Bootstrap data.

An arguably more appropriate monthly SIE estimate can be produced by instead using the monthly means of the daily SIEs -
as opposed to the calculation based on monthly mean SICs, described above (see, for example, Parkinson et al., 1999). As this is the approach used to produce the SIE values used and disseminated by scientists at the NASA Goddard Space Flight Center (GSFC) Cryospheric Sciences Laboratory (https://neptune.gsfc.nasa.gov/csb/index.php?section=234), we refer to this as the GSFC-SIE index. We thus also calculate the SIE (and SIA for consistency) to briefly explore the impact from this alternative methodology. Note that at the time of writing, the NSIDC are preparing to switch to this new methodology,
making a comparison of these different approaches timely.

We also use the monthly Arctic SIA and SIE to produce an estimate of sea ice compactness, which is simply the ratio of the total Arctic SIA/SIE (e.g. Comiso and Nishio, 2008, in which the ratio is referred to as concentration). Uncertainty surrounding the contribution of summer melt on the concentration estimates (e.g. melt ponds being flagged as open water) means less weight should be given to the summer (June-August) SIC and thus SIA and compactness estimates presented
later. As discussed earlier, this is thought to be less pertinent for the Bootstrap data, which use variable tie points, but is still likely to be significant.

### 2.2. Ice drift

Following Holland & Kimura (2016, referred to herein as HK2016) we use ice drift estimates to investigate the monthly concentration budgets of the Arctic sea ice pack (methodology discussed in the following section). The drift data are
produced from AMSR-E brightness temperatures from January 2003 to September 2011 and AMSR-2 brightness temperatures from July 2012 to December 2016 using a cross-correlation approach (see Kimura et al., 2013 for more details). Wintertime (January-March, November-December) ice drifts are derived using 36-GHz channels, while summertime drifts (April–October) are derived using 18-GHz channels, to maximize the reliability and coverage of the data. The data is provided at a 60 km x 60 km horizontal resolution. The drift data is referred to herein as KIMURA. Note that we
also explored sea ice drift estimates produced by the Centre ERS d'Archivage et de Traitement (CERSAT), part of the Institut Français de Recherché pour l'Exploitation de la Mer (IFREMER) (Girard-Ardhiun and Ezraty 2012), however the data had consistently lower coverage than the KIMURA dataset, likely due to the use of a stricter ice drift mask, limiting its utility for this study.

### 2.3. Melt and freeze onset

We use the timing of melt onset (MO) and freeze onset (FO) from NASA's Passive Microwave (PMW) MO and FO datasets



from 2000 to 2016, updated from Stroeve et al. (2014) and Markus et al. (2009). We use data regarding the date of 'continuous' MO, and 'late' FO from the PMW dataset. The continuous MO dates are consistent with the 'Melt Onset' transition periods defined by Livingstone et al., (1987). We compare the PMW MO estimates to MO data produced from the Advanced Horizontal Range Algorithm (AHRA) Snowmelt Onset on Arctic Sea Ice Version 3 product (Anderson et al.,

2014; Bliss and Anderson, 2014). The AHRA product provides the date of the earliest MO signal, consistent with the start of the Livingstone et al., (1987) 'Early Melt' season. The AHRA product is comparable to the PMW 'early' MO and is most consistent with the PMW 'early' MO, which is not used in this study as the AHRA is thought to be more sensitive to early melt transitions. Both PMW and AHRA datasets are based primarily on the sensitivity of $T_b$ to liquid water content in the overlying snow cover (see Bliss et al., 2017 for more detailed description of the two products and their differences). We

explore the MO/FO data within specific Arctic regions (as in Stroeve et al., 2014). We choose to focus our analysis on four different regions: the Central Arctic; the North Atlantic (defined by the Greenland and Barents seas); the Eastern Arctic (defined by the Kara, Laptev and East Siberian seas); and the Alaskan (Bering, Beaufort and Chukchi seas) regions. See Figure S2 for maps of these regions.

*2.4. Sea surface temperatures*

We use sea surface temperatures (SSTs) estimates from the National Oceanic and Atmospheric Administration (NOAA) Optimum Interpolation Sea Surface Temperature (OISST, version 2) data set, which is a daily, high-resolution (0.25° x 0.25°) dataset derived from a blend of *in-situ* observations and Advanced Very-High-Resolution Radiometer (AVHRR) satellite infrared data (Reynolds et al., 2007). Note that a comparison with ship-based CTD observations found a bias of only 0.02 °C, but a RMS error of 1.77 °C (Stroh et al., 2015).

*2.5. Atmospheric data*

Finally, we use daily sea level pressure and near-surface (10m) air temperature and wind speeds from the Modern-Era Retrospective Analysis for Research and Applications, version 2 (MERRA-2) reanalysis (Gelaro et al., 2017) to study the atmospheric conditions during the peak of the August 2016 (Aug 6[th]) and 2012 (Aug 16[th]) Arctic cyclones.

**3. Methods**

To explore the relative contribution of dynamic (e.g. ice export) and thermodynamic (i.e. melting/freezing) processes to Arctic sea ice variability, several studies have decomposed the ice concentration (or volume) budgets in either observations or model results (e.g., Lindsay and Zhang 2005, Holland et al. 2010, Holland & Kwok 2012, Holland, et al. 2014, Holland and Kimura 2016). Here we use the daily SIC and ice drift data to map the observed dynamic and thermodynamic budgets of Arctic SIC, following HK2016. The SIC and ice drift data are re-gridded onto the same 100 km polar stereographic grid

before the budget terms are calculated. A coarser spatial resolution than the drift data is used to reduce noise in the data




before the flux divergence term is calculated. The drift data are also smoothed using a Gaussian filter (as in Holland and Kimura, 2016). The monthly changes in SIC across the Arctic are decomposed into thermodynamics/dynamics (based on Eq. 1 and 2 in HK2016) as:

$$\frac{\partial A}{\partial t} + \nabla.\left(\boldsymbol{u_i}\,A\right) = R$$

where $\partial A/\partial t$ represents ice intensification (the change in SIC in a given grid cell over time) and $\nabla.\left(\boldsymbol{u_i}\,A\right)$ represents ice flux divergence (the change in SIC in a given grid cell from/to surrounding grid cells). The residual (R) on the right-hand side of Eq. 1 represents thermodynamic melting/freezing and mechanical redistribution (e.g. ridging and rafting), which should balance the total of intensification and flux divergence. A more detailed discussion of this concentration budget methodology is given in HK2016. Note that while it can be useful to separate the ice flux divergence term (change in SIC

driven by dynamics) into advection and divergence terms, and to present the residual as a separate term (as in HK2016), we avoid this extra step for simplicity, and instead focus on the ice intensification and flux divergence terms. Ice intensification and flux divergence are calculated daily, with intensification as a central difference in time, and flux divergence as the central difference in space. Both terms are then summed (monthly) from these quasi-daily estimates within each month. While the KIMURA ice drift data record contains gaps due to the AMSR-E/AMSR-2 operating periods (highlighted in the

previous section), we believe the data coverage is sufficient to represent a 'New Arctic' (2000-2015) climatology, from which we calculate the monthly 2016 flux divergence anomalies. Note also that the ice drift data are relatively uncertain compared to the SIC data, especially around the ice edge, meaning our ice flux divergence estimates are thought to be less reliable than the ice intensification estimates.

## 4 Results and discussion

**4.1 Monthly Arctic sea ice indices in 2016**

The monthly estimates of Arctic SIE calculated from the NASA Team and Bootstrap SIC data are shown as box and whisker plots in Figure 3 (2000-2015, 2016 indices highlighted by crosses). The mean seasonal cycle is clear in both datasets, with Arctic sea ice reaching its maximum (minimum) extent in March (September), as expected. The monthly SIE rankings are also given in Figure 3, with the NASA Team data indicating record low SIE in all months of the year except for summer

(July to September). It is interesting to note that the 2016 August and September SIE, the months which have seen the strongest long-term Arctic sea ice declines, are significantly above the previous monthly record lows, which were both set in 2012. As discussed earlier, the NSIDC Sea Ice Index SIE values show more significant differences in years prior to 2016 (albeit still < 0.05 million km$^2$) due, we believe, to the use of monthly SIC data in the NSIDC index, resulting in no record low 2016 SIE in March and December in that dataset. As is well established in the literature but worth repeating, record low

sea ice at the start of the year does not always translate to record low sea ice in summer, with the spring/summer weather conditions crucial in controlling the magnitude of seasonal ice loss.





Figure 3 also highlights the differences between the NASA Team and Bootstrap data, with higher SIEs calculated from the Bootstrap SIC data, as expected. The monthly SIEs produced from the two SIC products differ by around 0.2 to 0.5 million km$^2$. Similar to the NSIDC Sea Ice Index, the Bootstrap results indicate no record low 2016 SIE in March and December. The 2016 values are summarized in Table 1, including the values given by the NSIDC Sea Ice Index. Note that other studies
have provided a more in-depth assessment of the SIC algorithm differences (e.g., Comiso et al., 1997; Meier, 2005; Ivanova et al., 2015), including their impact on Arctic sea ice trends and variability (e.g. Comiso et al., 2017), so we focus instead on comparing the 2016 indices in context of the New Arctic regime (2000 onwards).

The monthly estimates of Arctic SIA are also shown in Figure 3 (and summarized in Table 1). As discussed earlier, we fill the pole hole for SIA using the mean SIC in a 0.5-degree halo around the pole hole (which the NSIDC does not do). The
values of SIA are lower than SIE, as should always be the case, and the differences between the monthly NASA Team and Bootstrap SIA indices are larger than SIE, as expected from previous studies comparing SIE and SIA across different algorithms (e.g. Comiso et al., 2017). The differences in monthly 2016 SIA calculated from the two SIC datasets are around 0.5 to 1.5 million km$^2$. The seasonal cycle of record low SIA is similar to SIE - record lows in winter/spring/autumn, but no record lows in summer, despite now factoring in the SIC within the ice pack. Similar to the SIE rankings, the choice of
algorithm determines how many monthly records were observed, with Bootstrap showing fewer records than NASA Team, in general (mainly at the start of the year). Figure 3 shows that the summer 2016 SIA values were closer to the record low values, however. The lack of a record low Bootstrap SIA in October was somewhat surprising considering the record low October SIE, implying that the 2016 October SIC was not particularly low compared to the 2000-2015 mean (explored more in the following compactness discussion). The November 2016 SIA still produces a clear record low, however, highlighting
the strong intra-seasonal variability in these indices. The record low SIA in December is noteworthy for its strong departure from the 2000-2015 spread, especially in the NASA Team data. In general, the 2016 November and December SIA indices show the biggest departures from the 2000-2015 distribution.

The differences between the Bootstrap and NASA Team indices are partly due to the different algorithm methodology (different channel combinations) and parameters (i.e., tie points for pure ice/water surface types). However, there are also
important differences in post-processing between the products. Both use weather filters to remove false ice in open water regions due to wind roughening of the ocean surface and precipitation, but each uses different approaches. Likewise, both use different methods to address land-spillover errors – false coastal ice due from mixed land/water in the sensor footprint. Finally, both products independently conduct a final manual quality-control procedure – removing retrievals considered to be in error; because it is manual, there is inherently some subjectivity in this procedure. The differences in these post-
processing steps can have differing impacts on the SIE and SIA from the two products at different times of year.

The SIE and SIA indices were also calculated using the monthly means of the daily SIE/SIA values (as opposed to using monthly mean SICs), as discussed earlier - the GSFC-SIE index. These 2016 results are also summarized in Table 1, with



the box and whisker plots shown in Figure S3. The differences just due to the different averaging methodology are significant, with SIE around 0.2 to 0.5 million km$^2$ lower across the two algorithms, and SIA around 0.05 to 0.01 million km$^2$ higher than the indices calculated using monthly SIC. For SIE, the choice of methodology results in differences as large as the difference caused by the choice of algorithm. The SIE values are lower using this method as the SICs below 0.15 are

removed each day (and thus the ice is not given a chance to increase to above 0.15 later in the month), increasing the amount of low SIC not included in the SIE calculations. While the use of daily means reduces the overall SIE values, it also impacts the rankings significantly, especially for the Bootstrap data. Now no record low Bootstrap SIE is indicated for January, February and October, but a record low is now indicated for December. The only impact on the NASA Team SIE rankings is the removal of the record low in October. The smaller impact of SIC averaging on SIA means that this change in

methodology only removes the record low October NASA Team-derived SIA. In summary, care must be taken when calculating and comparing SIE/SIA, especially for those concerned with estimating sea ice rankings and comparing across studies. We continue with the monthly mean SIC derived indices for the discussion of sea ice compactness below for simplicity.

Figure 4 shows box and whisker plots of sea ice compactness, *C*, the ratio of monthly pan-Arctic SIA/SIE. Note that this

approach was also presented and discussed in the 2016 post-season SIO report. The results demonstrate interesting similarities and differences between the two algorithms. The Bootstrap values of *C* are consistently higher than NASA Team, but the seasonal cycle is slightly damped. This was expected considering the low concentration bias in the NASA Team data, especially in summer. As noted earlier, the passive microwave sensor is sensitive to surface melt (although less so for the Bootstrap data) so the June-August data should be considered with caution. Both datasets show record low *C* in

September, but unexceptional behavior in January to July, and low *C* in August. Both also show record low *C* (although not as extreme as September) in November and December. The wide spread in the NASA Team *C* values in June may be due to the more significant inclusion of melt ponds in the concentration data (e.g. Kern et al., 2016) which peak in coverage through June. The extreme record low *C* index in September 2016 highlights the anomalous behavior of the ice pack in the summer of 2016, which we explore in more detail in the following sections. As discussed in the SIO, if the September 2016

Arctic ice pack had a more average *C* index, the observed September SIE could have been around 0.5 million km$^2$ lower, although still not low enough to set a new record low.

### 4.2 Budget analysis

Here we present and discuss the observed monthly sea ice concentration anomalies and concentration budgets, to explore the regional drivers of the monthly 2016 Arctic sea ice states.

### 4.2.1 Ice concentration and intensification

Figure 5 shows the monthly 2016 ice intensification, $\partial A/\partial t$, estimates, with the ice intensification anomalies shown in





Figure 6. Note that we show only the results produced using the Bootstrap SIC data for simplicity, but provide maps of the raw and anomaly ice intensification estimates produced using the NASA Team data in Figure S4 and S5. Note how the NASA Team intensification maps show more variability within the ice pack, especially in summer, which we believe may be influenced significantly by the changing surface conditions, hence our choice to focus more on the Bootstrap results. In

general, the seasonal variability in ice intensification is broadly in-line with the results shown in HK2016 (their Figure 4 shows 2003-2010 seasonal means), including mostly negative intensification in May-July, and positive intensification, in the peripheral Arctic seas, in November-January. The monthly maps across the entire year presented here provide further insight into the spatial variability of ice intensification (the summer of 2007 is also presented monthly in HK2016).

As presented earlier, the monthly 2016 SIC maps are shown in Figure 1, with the SIC anomalies (compared to the 2000-

2015 mean) shown in Figure 2. Note that Figure 2, 5 and 6 also include the location of the monthly 2016 sea ice edge, calculated using the 0.15 SIC contour. We discuss the results below by season and focus primarily on the anomaly maps (Figures 2 and 6).

*Winter (January-March):* The winter SIC anomalies show a bimodal pattern of negative anomalies in the Barents, Kara and Bering seas, which drove the record low winter SIE/SIA, and positive anomalies in the Labrador Sea and the Sea of

Okhotsk. These latter regions appear to have prevented the SIE from reaching even lower record values in winter 2016. As discussed in Boisvert et al*.,* (2016), an extreme winter cyclone caused significant sea ice declines in the Barents and Kara seas at the start of 2016, followed by a slower increase in SIC through the middle/end of January (see their Figure 6a). This low SIC state in the Barents and Kara seas persisted through the winter season. The pattern of intensification anomalies are more variable, and primarily highlight regions adjacent to the ice edge that experienced strong increases/decreases in SIC

due to the anomalous location of the ice edge during that month. For example, while the Bering Sea shows negative SIC anomalies through winter, the intensification (and anomaly) is positive in some regions south of the ice edge in March, as the ice advance occurred later than usual in this region. In January we see small regions of moderate and positive intensification anomalies in the Barents and Kara seas and the Sea of Okhotsk. The positive (negative) intensification anomaly in the Sea of Okhotsk, and to a lesser extent in the Labrador Sea, in January/February (March) correspond with delayed freeze-up in these

more southerly Arctic regions.

*Spring (April-June):* The April results show a similar spatial pattern of SIC anomalies, including persistence of the low SIC state in the Barents and Kara seas. The negative SIC anomaly in the Sea of Okhotsk persisted until April, while the Labrador Sea SIC anomaly extended westward into Baffin Bay and Hudson Bay. The Bering Sea shows negative (positive) intensification anomalies in April (May), due to the earlier ice retreat in the region. The strongest SIC anomalies are

observed in the southeastern Beaufort Sea in May and June, which are associated with negative intensification anomalies in April, followed by positive intensification anomalies in June (the SIC cannot decline any further). Some positive intensification anomalies are present within the Central Arctic, north of the ice edge, highlighting some regions where the




loss of SIC was slower than normal. Note that we explore the regional melt onset in more detail later (Section 4.4).

*Summer (July-September):* The summer results feature interesting spatial patterns of SIC anomalies within the central Arctic Ocean, including positive (negative) SIC anomalies in the Laptev and Chukchi (Beaufort and East Siberian) seas. The intensification anomalies instead feature a bimodal temporal pattern of positive (negative) intensification anomalies in the

Eastern (Western) Central Artic in August (September). The strong positive anomaly in the eastern central Arctic appears to have contributed significantly to the lack of a record low 2016 September SIE/SIA. Indeed a rapid increase in SIC following the daily minimum SIE (recorded on September 10th) was highlighted at the time by the NSIDC (http://nsidc.org/arcticseaicenews/2016/10/). It appears that the negative intensification anomalies in August were not strong enough to increase SSTs sufficiently to prevent the relatively rapid recovery of the ice pack in this region through

September. We explore the SST response in Section 4.3.

*Autumn (October-December):* The autumn SIC anomalies are mainly negative, but include some small regions of positive SIC anomalies in the Laptev Sea (in October) and the Labrador Sea and the Sea of Okhotsk (in December). It is interesting to note the similarity in the January and December SIC and anomaly maps (the year started and ended in a similar state). The intensification anomalies (and raw fields) through autumn (and also September) appear to be generally stronger than in the

other seasons. The negative intensification anomalies in October throughout the peripheral Arctic seas highlight the delayed October refreeze of the Arctic Ocean in 2016 (we present and discuss freeze onset in the following section). The negative October intensification anomalies are followed by positive intensification anomalies in November, as the sea ice refreeze began later than expected in the Beaufort, Chukchi, East Siberian and southern Kara seas. In December, the lack of refreeze in the Bering and Barents seas are associated with negative intensification anomalies in this region, and the significant areas

of low SIC ice that remained at the end of the year (Figure 1), contributing to the record low SIA index discussed in the previous section.

We also analyzed anomaly fields prior to 2016, to assess when the SIC anomalies appeared. Maps of the SIC and SST anomalies for September to December 2015 are shown in Figure S6. The SIC anomaly maps show that the Barents Sea SIC anomalies appeared as early as October 2015 and persisted into, and through, 2016. The negative (Sea of Okhotsk)

anomalies appeared in December 2015, while the positive (Labrador Sea) anomalies appeared in November and persisted through December.

### 4.2.1 Flux divergence

An additional driver of the 2016 SIC anomalies is from ice dynamics - the combination of ice divergence/converge and advection (ice drift combined with spatial gradients in SIC). Figure 7 shows the monthly 2016 ice flux divergence,

$\nabla.(\boldsymbol{u_i} A)$, anomalies. Note that negative values of the flux divergence correspond to 'dynamical' ice loss, and vice versa. The winter results show a combination of positive and negative flux divergence anomalies, including some anomalous



dynamical ice loss in Hudson Bay (in January) and the southeastern Beaufort Sea and Kara Sea (in February), but anomalous dynamical ice gain in the seas north of Svalbard (in February and March). Some anomalous dynamical ice loss is indicated in the southeastern Beaufort Sea in April, potentially helping precondition the region for the strong SIC declines observed in May and June. The timing of ice retreat in this region is thought to be increasingly important in controlling the total ice loss

through summer (e.g. Steele et al., 2015). The maps of ice drift (Figure 1) show this was associated with a strong Beaufort Gyre ice circulation, which has been strengthening over recent decades (Petty et al., 2016). No obvious spatial patterns are observed in the May-June maps; however the flux divergence anomalies appear stronger in August onwards. The summer results show a similar (albeit less obvious) bimodal pattern of anomalous dynamical ice loss (gain) in August (September) in the central Arctic. The strongest anomalies are observed in October, including strong anomalous dynamical ice gain along

the Siberian coastline and an associated (but weaker) dynamical ice loss in the northern Beaufort/Chukchi seas. This dynamical ice loss from the Beaufort/Chukchi seas appears to have helped drive the record low October SIE as the corresponding SIC gains along the Siberian coastline due to this drift circulation could not increase the extent of the sea ice pack. November and December show in general more regions of anomalous dynamical ice gain than loss, meaning ice dynamics are not thought to have been a significant contributor to the record low late autumn sea ice states.

**4.3 The melt season and sea surface temperatures**

Figures 8 and 9 show the Arctic sea ice melt onset (MO) and freeze onset (FO) respectively, from 2000-2016 for four different Arctic regions (data described in Section 2, region maps shown in Figure S2). The MO and FO are presented as anomalies relative to the 2000-2016 mean. Note that Figure 8 shows the MO data from both the NASA PMW and AHRA MO products. In general there is good agreement in the interannual variability between the two MO products, including a

general trend towards earlier MO throughout the 2000-2016 period. While differences between the AHRA and PMW MO are expected (the PMW data used indicate the start of continuous sea ice melt, while the AHRA is capturing the early MO period), both MO estimates show positive anomalies (indicating earlier MO) in our four Arctic regions in 2016. Note that as we include open water in our calculations (set to a constant of day 61, the earliest MO date) the sensitivity of these regional means to the coverage of open water is likely to be significant. We decided on this approach, as opposed to simply masking

the open water, as we wanted to include open water in our calculations to give a more consistent metric of sea ice melt. While the interannual variability in the North Atlantic MO anomaly is small, due in part to the low coverage of sea ice in this region relative to open water, we do observe an anomalously early MO date in the region in 2016 in both the PMW and AHRA data. The AHRA data also indicate an anomalously early 2016 MO date in the Alaskan (Bering, Beaufort, Chukchi seas) region, although both MO products show that the 2016 Alaskan MO date continues the trend of early MO dates set in

2014 and 2015.

The 2016 PMW FO anomalies (Figure 9) show higher interannual variability than the MO data (note the different scales on the y-axes). The 2016 results show record late dates in the Central Arctic, North Atlantic and especially the Alaskan regions.



The Alaskan FO is around 10 days later than the next record high FO, which was set in 2007. The Eastern Arctic FO 2016 anomalies are similar to the highs indicated in 2007, 2011 and 2012, with these results pointing more towards a step-change in FO since 2007 (albeit with the potential for earlier FO to return, as indicated in 2013).

In Figure 10 we show monthly maps of the NOAA SST data, to briefly highlight and explore the link between the 2016 sea
ice cover, SSTs and MO/FO. In general the SSTs are ~2-3 °C warmer in the North Atlantic and Barents sea regions from January onwards. SST anomalies persist throughout the year in the Barents Sea, which peak in the late spring-early autumn, including small regions of SST anomalies over 5 °C in July in the southern Barents Sea. There is a small relative decrease in SST in August, which may be associated with the strong cyclones that entered the Arctic during this time (discussed more in the following section). The Kara Sea SST anomalies appear in June and generally persist until October, although they did
decrease to mean values in August. The SST anomalies start to appear in the southeastern Beaufort Sea in April onwards, when we observed the anomalous dynamical ice loss in the region. The June SST anomalies in this region are up to ~5 °C higher than the mean. The strong temperature anomalies in October and, to a lesser extent, November, in the Barents Sea and the Bering/southern Chukchi seas, appear to have been crucial in delaying ice freeze-up.

As in the SIC anomaly discussion, we also assessed the SST anomalies for several months prior to 2016 (September 2015
onwards), as shown in Figure S6. These indicate that the Barents Sea SST anomalies appeared as early as September 2015 and persisted through until 2016. However, these anomalies were not as strong as the September-December SST anomalies observed in 2016.

### 4.4 The summer Arctic cyclones of 2016

The previous sections highlighted the summer (August and September) as a particularly interesting time period in the 2016
Arctic sea ice annual cycle. As discussed by the NSIDC (http://nsidc.org/arcticseaicenews/2016/09/arctic-sea-ice-nears-its-minimum-extent-for-the-year/) and a more detailed study by Yamagami et al., (2017), August 2016 featured two cyclones that entered the Arctic Ocean. The first cyclone was fed by anomalously warm and moist air over the Barents Sea and warm air over northwestern Siberia. The cyclone's central pressure dropped to 968 hPa on August 16[th], while on August 22[nd], a second storm moved into the central Arctic Ocean along a similar track, and on August 23[rd], attained a central pressure of
970 hPa. Note that the total SIE decline during August 2016 was 2.34 x 10$^6$ km$^2$ (based on NASA Team data). The storm resulted in strong winds in excess of 22 m/s and waves as high as 4 meters along the ice edge in the East Siberian Sea. Waves from strong cyclones act to break up the ice cover and mix SSTs with warmer water below and thus have the potential to enhance basal and lateral ice melt. The sea level pressure, near-surface winds, near-surface air temperature and temperature anomalies during the peak of this storm period using NASA's MERRA-2 reanalysis data are shown in Figure
30  10.

It is interesting to compare the impact of these cyclones to "The Great Arctic Cyclone of August 2012" (Simmonds and



Rudeva, 2012). The sea level pressure, near-surface winds, air temperature and temperature anomaly during the peak of this storm are also shown in Figure 10. The 2012 cyclone entered the Arctic Ocean from Siberia in August 6[th], and traveled into the Chukchi Sea. The central pressure dropped to 966 hPa, the lowest recorded during the satellite data record (Simmonds and Rudeva, 2012), and remained below 1000 hPa for 10 days. While cyclones are generally associated with cooler

temperatures and ice divergence, the ice extent dropped by 2.72 x 10$^6$ km$^2$ during August 2012 (compared to 2.34 x 10$^6$ km in 2016), leading to a new record low for the month of September at 3.62 x 10$^6$ km$^2$ (Parkinson and Comiso 2013). This was the largest amount of ice lost during the month of August since at least 1979, higher than the observed ice loss in August 2016. While a new record low would have likely occurred regardless of the storm (Zhang et al. 2013), the timing of the storm (in August rather than in June) and relatively thin ice, resulted in fast removal of ice by increased mixing in the

oceanic boundary layer and advection of ice into warmer waters (Zhang et al., 2013).

In contrast to the 2012 August cyclone, which had its main centre of action in the Chukchi Sea, the cyclones in 2016 were located at the boundary between relatively thick ice north of the Canadian Archipelago and thinner ice in the East Siberian Sea, which may also have reduced their impact. The MERRA-2 data shown in Figure 11 suggests the 2012 storm centre experienced near surface air temperatures ~2 $^o$C warmer than the 2000-2016 mean, whereas the 2016 storm centre

experienced temperatures ~2 $^o$C cooler, which could also have contributed to the decreased ice loss in 2016. Such weather events are unpredictable on seasonal time scales and will thus always provide some limit to the skill and accuracy of summer Arctic sea ice forecasts, as we discuss later. Understanding their potential impact, however, could help us understand how big this barrier might be, and if it could change in the future.

**4.5 Implications for Arctic sea ice forecasting**

As discussed in Petty et al., (2017), the unconsolidated summer 2016 ice cover posed a challenge for those forecasting Arctic September SIE. Indeed while the forecasts presented in Petty et al., (2017) performed especially well over the last several years, the three forecast models utilized in that study (using SIC, MO and simulated melt pond coverage data, see Petty et al., 2017 for more details) all failed to accurately forecast the 2016 September Arctic SIE. Similar to 2012, the July SIO median forecast for the 2012 September SIE was biased high.

Here we briefly explore the potential improvements in forecast skill from forecasting September Arctic SIA, as opposed to SIE, considering the anomalously low compactness of the summer 2016 sea ice cover. We show only the SIC derived forecasts, as these produced the most skilful June (seasonal time-scale) forecasts of September SIE, especially when judged over recent years (since 2008). As discussed above, we use our own index of September SIA by filling in the variable pole hole, which have been used in these updated forecasts (we use the NASA Team data here). Again it is worth noting that for a

more thorough longer-term assessment, more sophisticated methods may be more appropriate, e.g. interpolating SIC data across the pole hole (Strong and Golden, 2016).

The SIC derived forecasts of September SIE and SIA are shown in Figure 12. The SIA forecast skill assessed for the 2008-2016 forecasts is higher (S=0.64) than the SIE forecast skill (S=0.56), which is largely, but not fully, driven by the improved



accuracy of the September 2016 SIA forecast. This simple comparison suggests that forecasts of September Arctic SIA could be more skilful than forecasts of SIE, especially in years that experience a more unconsolidated (lower compactness) summer ice cover, as in 2016. Put another way, this suggests it might be easier to predict how much ice there is, compared to the distribution/consolidation of the ice pack, as the latter is controlled more by unpredictable summer weather events. We

hope to explore this more in future work, especially as we move towards stakeholder focussed forecasts of Arctic sea ice (e.g. specific regions) and also months other than September.

**Summary**

In this study we explored the 2016 Arctic sea ice cover in terms of its monthly SIE and area (SIA), placing this in context of the sea ice conditions observed since 2000. We sought to highlight if and when monthly record low sea ice states were

observed in 2016, and the processes that contributed to this seasonal variability. The monthly 2016 SIE estimates used in the study were produced using two widely used daily sea ice concentration (SIC) datasets, the NASA Team and Bootstrap datasets, which resulted in differences in monthly SIE of around 0.2 to 0.5 million km$^2$, with Bootstrap consistently higher than NASA Team, as expected. The monthly Bootstrap SIA estimates, calculated in this study using the daily SIC data and filling the pole hole, showed even higher differences (Bootstrap estimates ~0.5 to 1.5 million km$^2$ higher). In general, fewer

monthly record lows were observed in 2016 when using the Bootstrap SIC data, especially in the early winter months. We also demonstrated that calculating monthly SIE/SIA from the monthly average of daily SIE or SIA estimates, instead of using monthly SIC data has a significant impact, of similar magnitude to the algorithm difference (differences in SIE of up to 0.5 million km$^2$), which also had a significant impact on the 2016 rankings (less records in 2016 in the Bootstrap data). Despite these differences, no combination of SIC data product or methodology resulted in a record low SIE or SIA in July,

August or September of 2016.

We also used the monthly SIA estimates to analyse the relative 'compactness' of the Arctic sea ice cover, the ratio of sea ice area over extent, highlighting anomalously/record low ice compactness in the summer of 2016, which helped contribute to the higher than forecast September SIE (from Petty et al., 2017 and the forecasts summarized in the 2016 Sea Ice Outlook).

Two cyclones that entered the Arctic Ocean during August appear to have contributed to the low SIC/compactness sea ice cover, but were not sufficient to cause more significant melt out and a new record low September SIE. A combination of colder temperatures and differences in storm track compared to the summer 2012 Arctic cyclone appear to have reduced the resultant ice loss. The implicit detection/inclusion of surface melt in the passive microwave data make the summer SIC estimates uncertain, however, especially for the NASA Team data. While the SIE/SIA indices provide a useful tool for

indicating the state of the Arctic sea ice system, care must be taken when considering what these indices mean, and how they are calculated. We highlight the conversion of SIC to SIE/SIA as arguably an overlooked issue, to-date, and something worth considering as the NSIDC transitions towards this methodology. Sea ice area, although a more uncertain variable, may offer potential benefits for those interested in producing accurate sea ice forecasts.



A concentration budget analysis was used to explore the regions and processes (thermodynamics/dynamics) contributing to, and indeed responding to the monthly 2016 sea ice conditions. In winter, the low to record low sea ice states were driven by low SIC in the Barents, Kara and Bering seas, with the ice intensification anomalies highlighting regions contributing to and

responding to the anomalous location of the sea ice edge. SIC anomaly maps show that the Barents Sea SIC anomalies appeared as early as October 2015 and persisted into, and through, 2016, contributing to the record early melt onset in the North Atlantic sector of the Arctic Ocean. Strong negative SIC and intensification anomalies, and positive flux divergence anomalies, appeared in the southwestern Beaufort Sea in spring. Summer featured interesting bimodal patterns of SIC and intensification anomalies, with the strong positive intensification anomaly in the eastern central Arctic through September

contributing to the lack of a record low 2016 September SIE/SIA. Freeze onset data show record late 2016 freeze up in the Central Arctic, North Atlantic and Alaskan Arctic region in particular, associated with strong sea surface temperature anomalies that appeared shortly after the 2016 minimum (October onwards), contributing to the return of record low SIE/SIA through the end of 2016.

The relative role of preconditioning, seasonal atmospheric/ocean forcing, and storm activity in determining the evolution of the Arctic sea ice cover is still highly uncertain, and worthy of more attention as we look to increase our ability to predict and understand the future evolution of the Arctic sea ice pack.

**Code availability**

After completion of peer review we will be including a link to the Python scripts used to generate the sea ice indices and concentration budgets presented in the this study.

**Data availability**

The sea ice concentration data are made available through the NSIDC, including the 2000-2016 NASA Team (http://nsidc.org/data/nsidc-0051) and 2000-2015 Bootstrap (http://nsidc.org/data/docs/daac/nsidc0079_bootstrap_seaice.gd.html) data. The 2016 Bootstrap data were provided by J. Comiso and will be archived at the NSIDC. The NSIDC Sea Ice Index can be accessed at

https://nsidc.org/data/seaice_index/. The PMW Melt Onset data are available through NASA's Cryospheric Sciences homepage (http://neptune.gsfc.nasa.gov/csb/index.php?section=54), while the AHRA Melt Onset data are made available through the NSIDC (http://nsidc.org/data/docs/daac/nsidc0105_arctic_snowmelt_onset_dates.gd.html). The KIMURA drift data are available by N. Kimura on request. The sea ice indices and concentration budgets will be made available by the author after completion of peer review.




**Acknowledgements**

We would like to thank the NASA Goddard Space Flight Center, Cryospheric Sciences sea ice group for inspiring conversations regarding the data and methods used in this study. We also thank Joey Comiso and Robert Gersten for providing the 2016 Bootstrap concentration data.

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

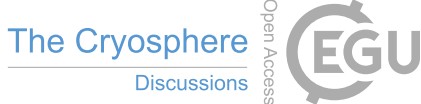



**Table 1**: Monthly 2016 Arctic sea ice extent (SIE) and sea ice area (SIA) calculated using the NASA Team and Bootstrap sea ice concentration data, along with the SIE/SIA given by the NSIDC Sea Ice Index (v2.1). The bottom rows (daily data)
10    show the values calculated using monthly means of daily SIE/SIA values. Values in bold indicate a new record low.

|  | Jan | Feb | Mar | Apr | May | Jun | Jul | Aug | Sep | Oct | Nov | Dec |
|---|---|---|---|---|---|---|---|---|---|---|---|---|
| **SIE** | | | | | | | | | | | | |
| Bootstrap | **13.83** | **14.52** | 14.77 | **13.95** | **12.28** | **10.88** | 8.71 | 6.13 | 5.26 | **6.91** | **9.48** | 12.51 |
| NASA Team | **13.63** | **14.32** | **14.52** | **13.82** | **12.07** | **10.60** | 8.12 | 5.59 | 4.71 | **6.44** | **9.07** | **12.08** |
| NSIDC (NASA Team) | **13.64** | **14.32** | 14.53 | **13.83** | **12.08** | **10.60** | 8.13 | 5.60 | 4.72 | **6.45** | **9.08** | 12.09 |
| **SIE** *(daily means)* | | | | | | | | | | | | |
| Bootstrap | 13.62 | 14.38 | 14.59 | **13.72** | 11.97 | 10.62 | 8.28 | 5.77 | 4.95 | 6.39 | **9.02** | **11.88** |
| NASA Team | **13.46** | 14.20 | 14.39 | 13.68 | 11.89 | 10.34 | 7.89 | 5.38 | 4.48 | 6.03 | **8.63** | **11.46** |
| **SIA** | | | | | | | | | | | | |
| Bootstrap | 12.68 | **13.35** | 13.35 | 12.68 | 10.89 | **9.11** | 6.75 | 4.39 | 3.96 | 5.59 | **8.15** | 10.76 |
| NASA Team | **11.70** | **12.31** | **12.51** | **11.92** | **10.16** | 8.11 | 5.26 | 3.23 | 2.82 | **4.29** | **6.94** | **9.56** |
| NSIDC | 11.68 | **12.29** | 12.48 | 11.89 | **10.14** | 8.09 | 5.24 | 3.21 | 2.81 | 4.27 | **6.92** | **9.54** |
| **SIA** *(daily means)* | | | | | | | | | | | | |
| Bootstrap | 12.73 | **13.40** | 13.58 | **12.73** | 10.96 | **9.16** | 6.82 | 4.45 | 4.00 | 5.65 | **8.23** | 10.81 |
| NASA Team | **11.73** | **12.34** | 12.54 | 11.95 | 10.21 | **8.15** | 5.33 | 3.30 | 2.86 | **4.35** | **6.99** | 9.60 |





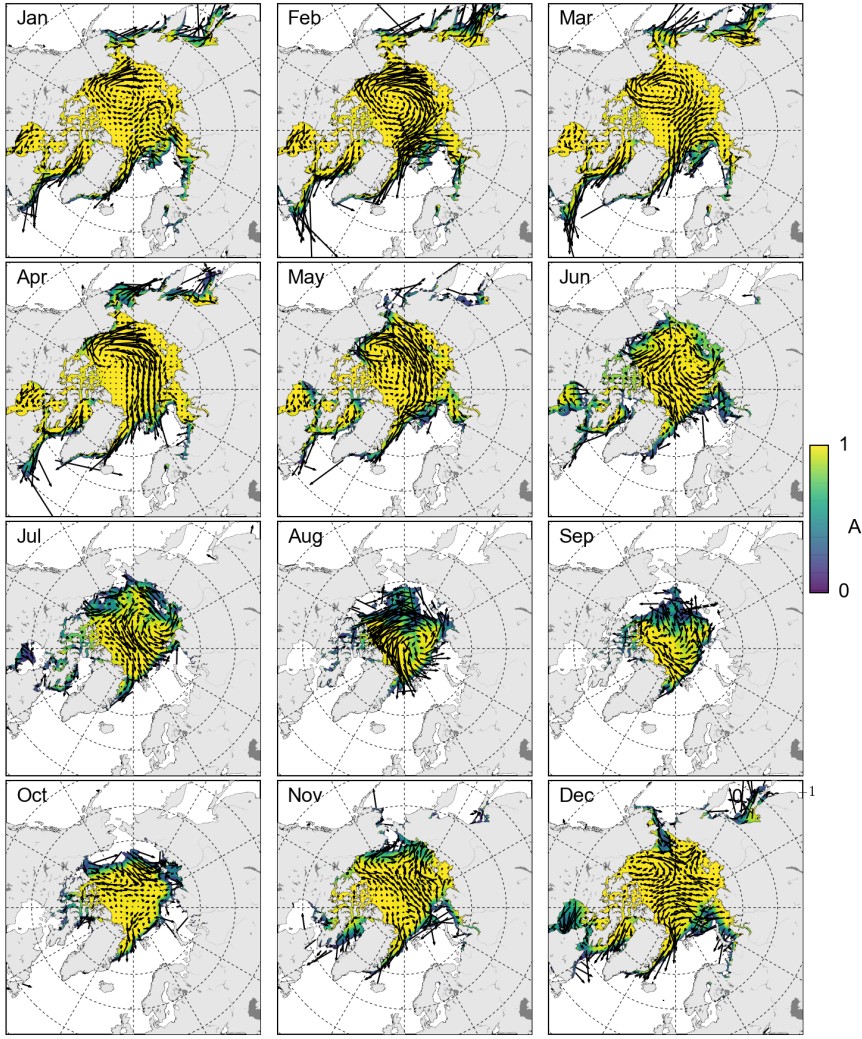

**Figure 1**: Monthly 2016 Arctic sea ice concentration using the Bootstrap algorithm, overlaid with the monthly mean KIMURA ice drift vectors (every third drift vector shown). The concentrations below 0.15 have been masked.



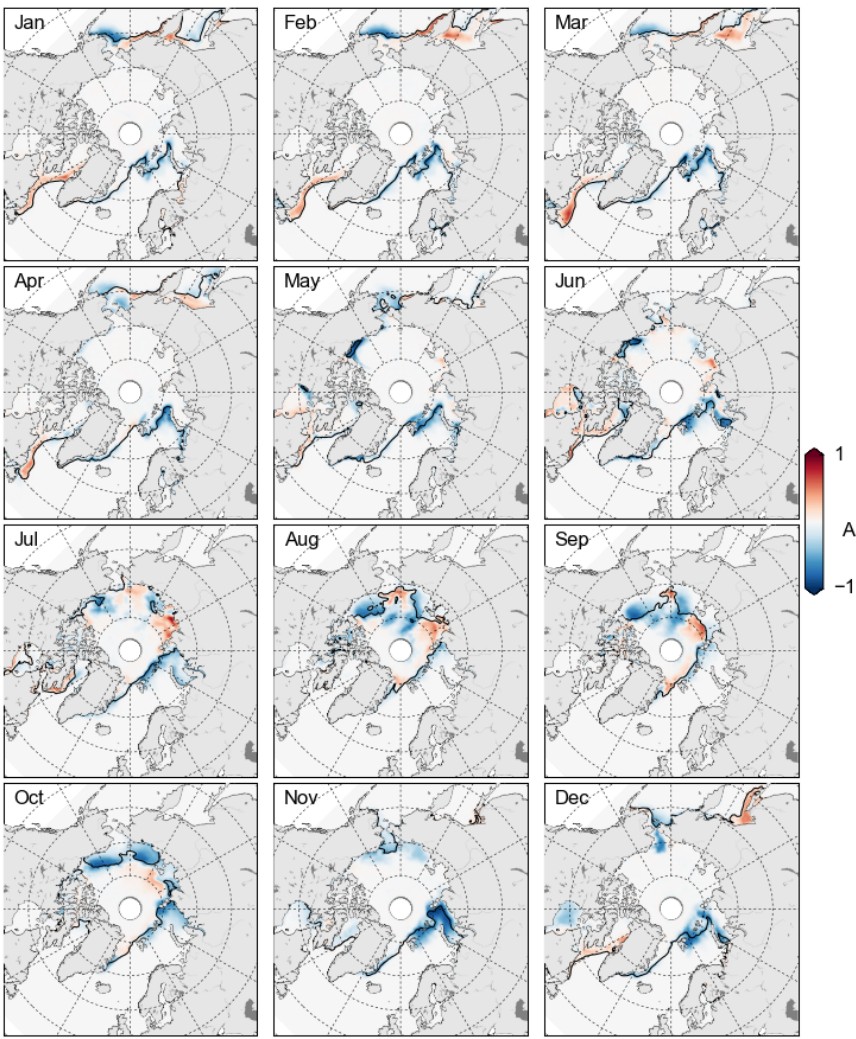

**Figure 2**: Monthly 2016 sea ice concentration (SIC) anomalies, relative to the 2000-2015 mean, using the Bootstrap algorithm. Note that the SIC data north of 86.5 °N is masked due to the pole hole present prior to 2008. The black contour indicates the monthly 2016 sea ice edge using a 0.15 SIC contour.





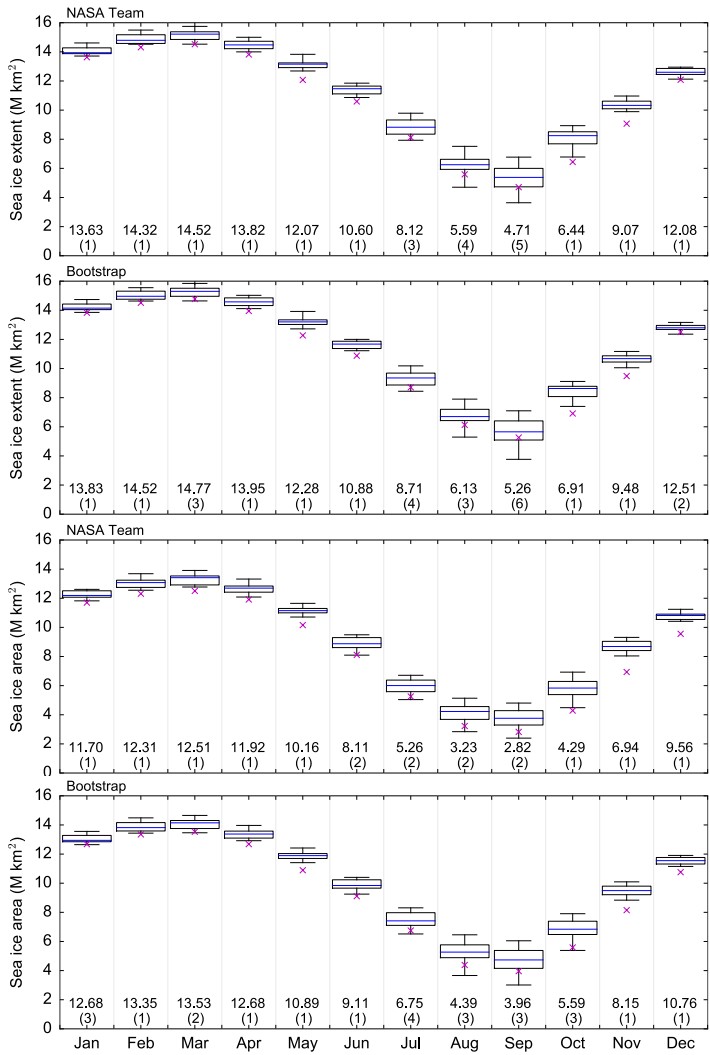

**Figure 3:** Box and whisker plots of observed monthly sea ice extent (SIE) for the period 2000-2015 calculated using the NASA Team (top), and Bootstrap (bottom) sea ice concentration data. The magenta crosses and the number above the brackets (in million km$^2$) denote the monthly 2016 SIE, while the number in brackets gives the rank of the 2016 SIE across the 2000-2016 period (1 = a record low in 2016).





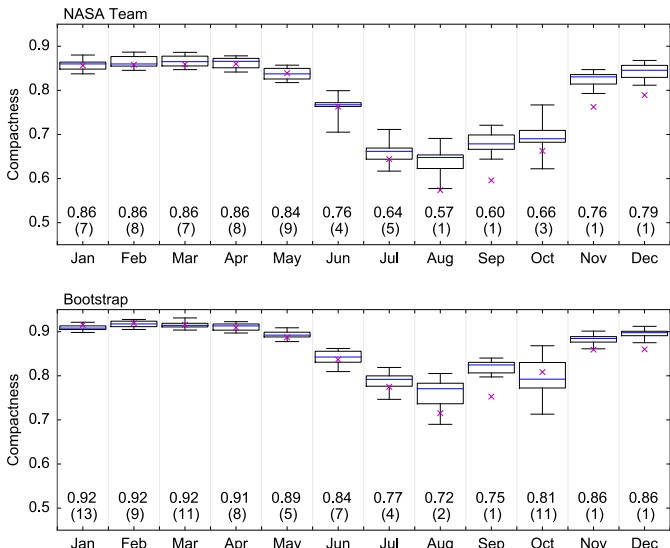

**Figure 4:** Arctic sea ice compactness from 2000 to 2015, calculated as the monthly Arctic sea ice area (SIA) over sea ice
10    extent (SIE), using the NASA Team (top) and Bootstrap (bottom) data. The magenta crosses and the number above the
brackets denote the monthly 2016 SIA/SIE, while the number in brackets gives the rank of the 2016 compactness across the
2000-2016 period (1 = a record low in 2016).





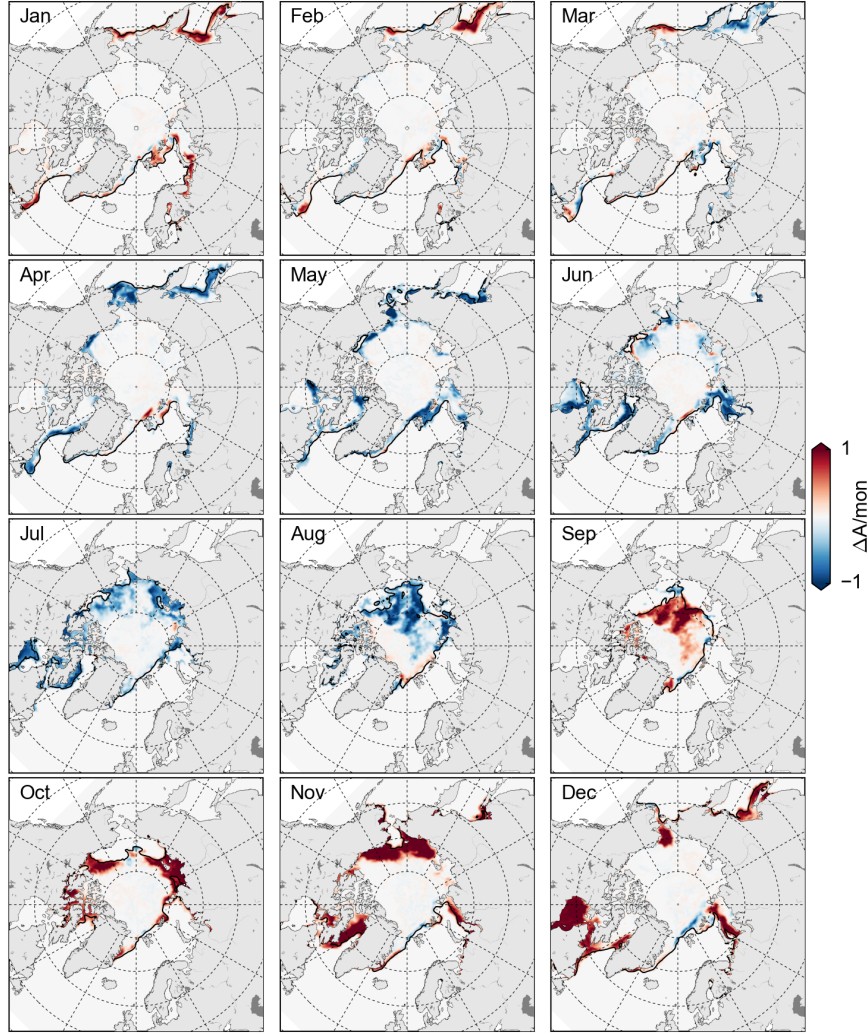

**Figure 5**: Monthly 2016 Arctic sea ice intensification estimates, calculated using the daily Bootstrap sea ice concentration data. Positive values (red) denote ice gain in a given grid cell. The units are concentration per month. The magenta contour indicates the monthly 2016 sea ice edge (0.15 ice concentration contour).




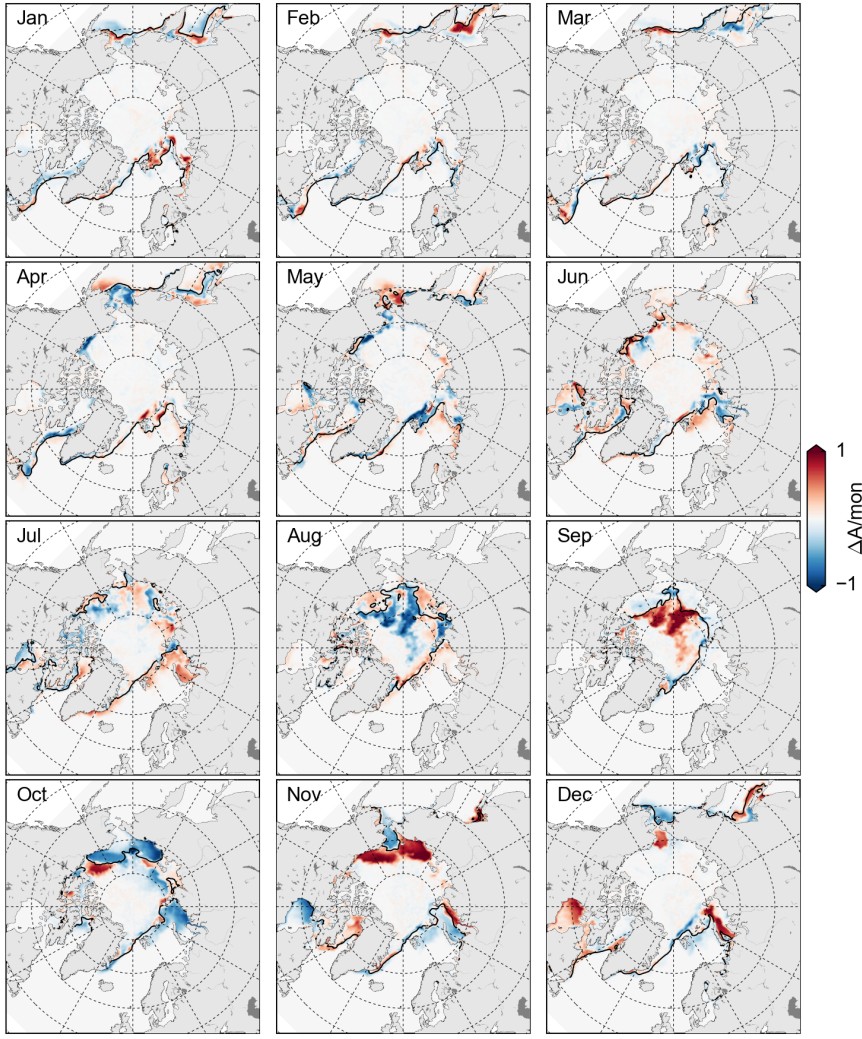

**Figure 6**: Monthly 2016 sea ice intensification anomalies, relative to the 2000-2015 mean, calculated using the daily Bootstrap sea ice concentration data. Positive values (red) denote more ice gain in a given grid-cell compared to the mean. The units are concentration per month. The black contour indicates the monthly 2016 sea ice edge (0.15 ice concentration contour).





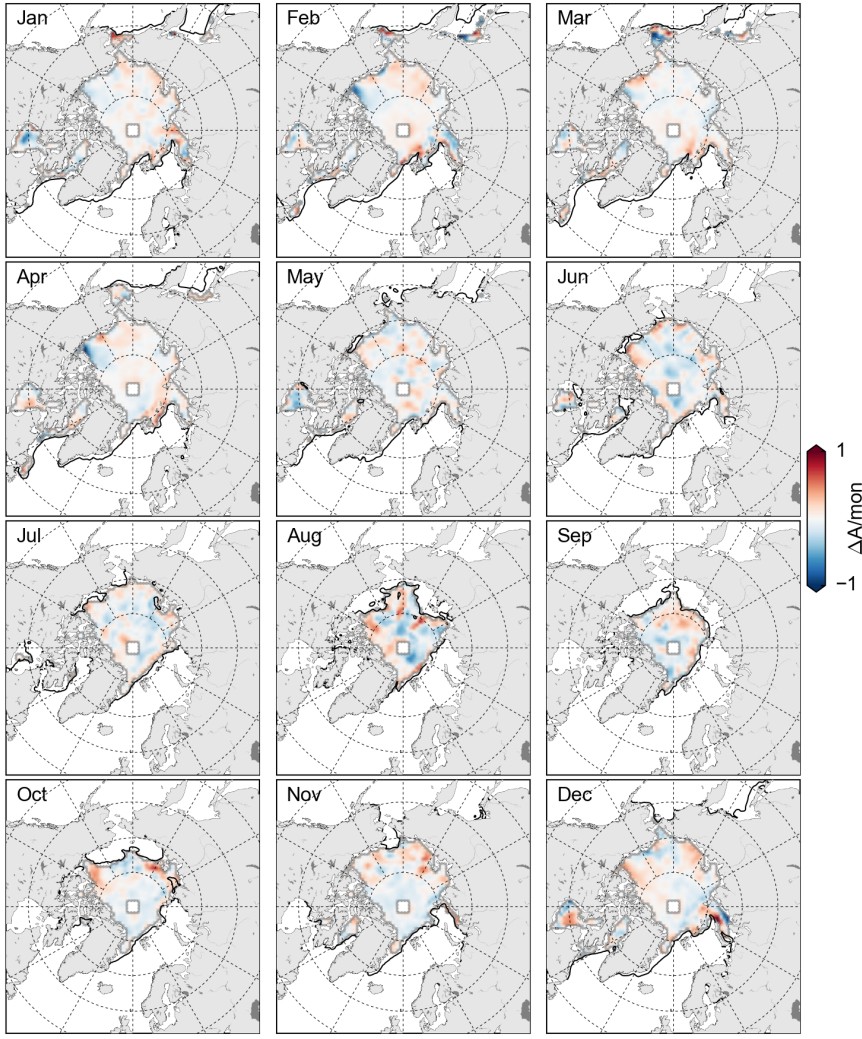

**Figure 7**: Monthly 2016 flux divergence anomalies, relative to the 2003-2015 mean, calculated using the daily Bootstrap sea ice concentration data and Kimura drift data. Positive values (red) denote more ice gain in a given grid-cell compared to the mean. The units are concentration per month. The black contour indicates the monthly 2016 sea ice edge (0.15 ice concentration contour).





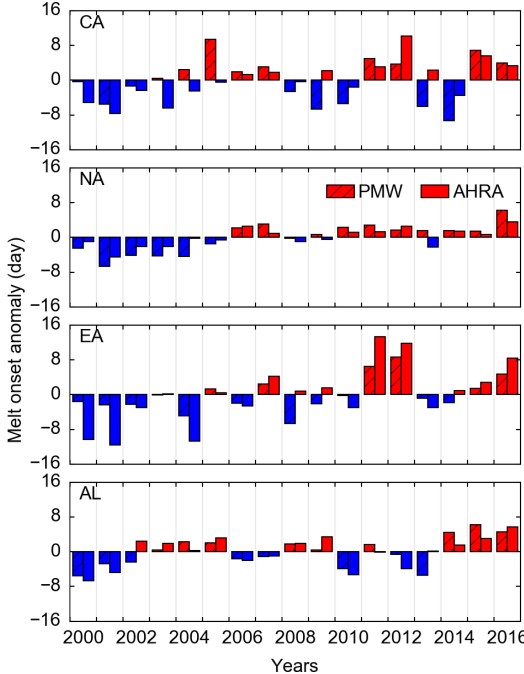

**Figure 8**: Melt onset (MO) date for four different Arctic regions using the NASA PMW late MO data and the AHRA MO estimates. All data are presented as anomalies relative to the 2000-2016 mean, with a positive (negative) value and red (blue) bars indicating an earlier (later) MO date. The regions (top to bottom) include the: Central Arctic (CA); North Atlantic (NA); Eastern Arctic (EA) and Alaskan (AL), regions, shown in Figure S2.



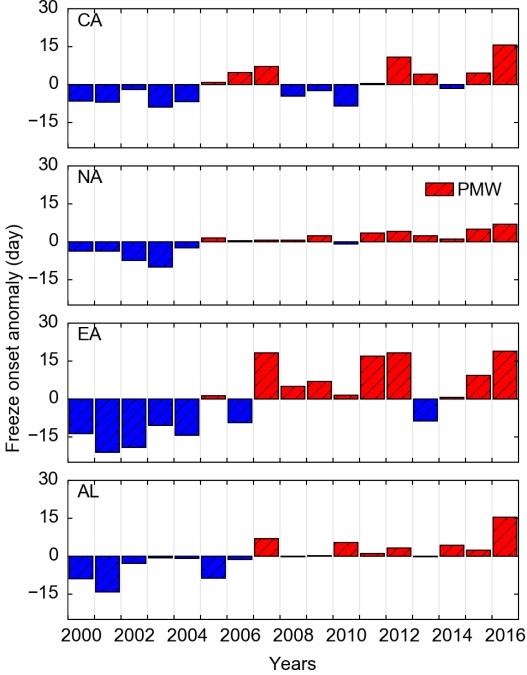

**Figure 9**: Freeze onset date for four different Arctic regions using the NASA PMW late freeze onset estimates. All data are presented as anomalies relative to the 2000-2016 mean, with a positive (negative) value and red (blue) bars indicating a later (earlier) FO date. The regions (top to bottom) include the: Central Arctic (CA); North Atlantic (NA); Eastern Arctic (EA), and Alaskan (AL), regions, shown in Figure S2.



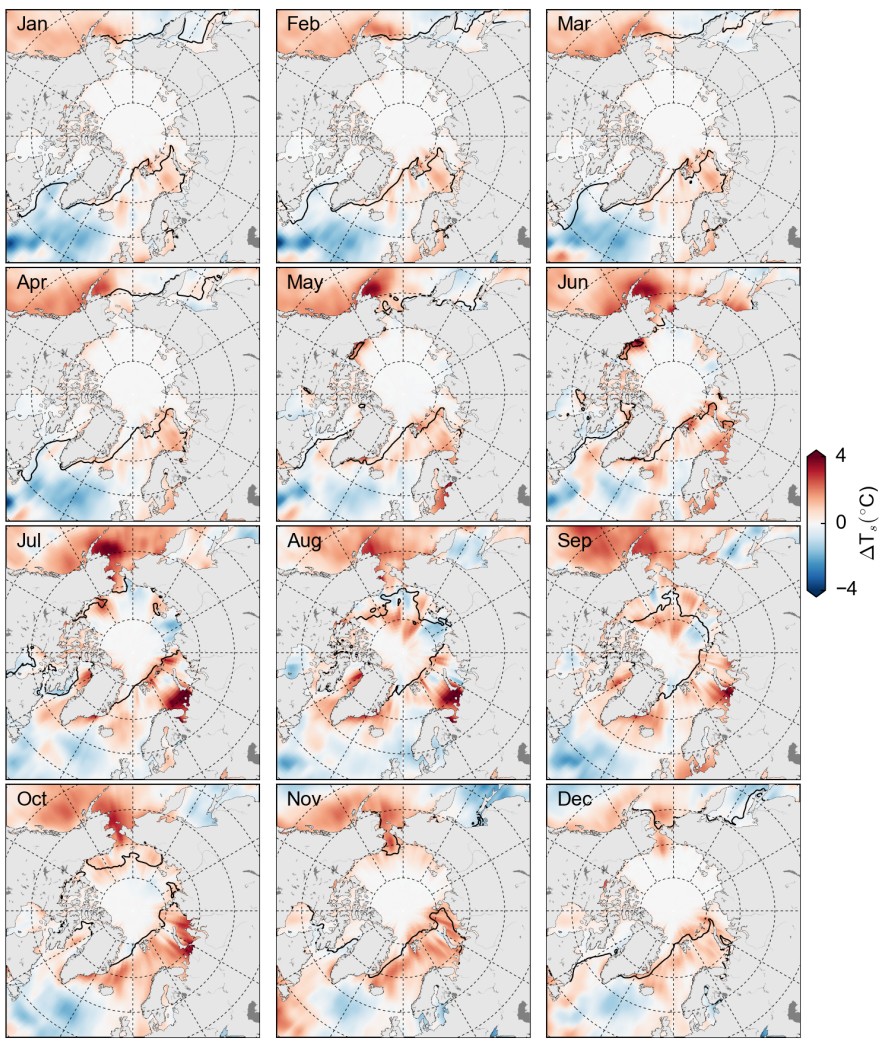

**Figure 10**: Monthly 2016 sea surface temperature (SST) anomalies (relative to the 2000-2015 mean) from NOAA's OISST dataset. 2016 raw SST shown in Figure S3. Add in SIE line (magenta). The black contour indicates the monthly 2016 sea ice edge (15% concentration contour).





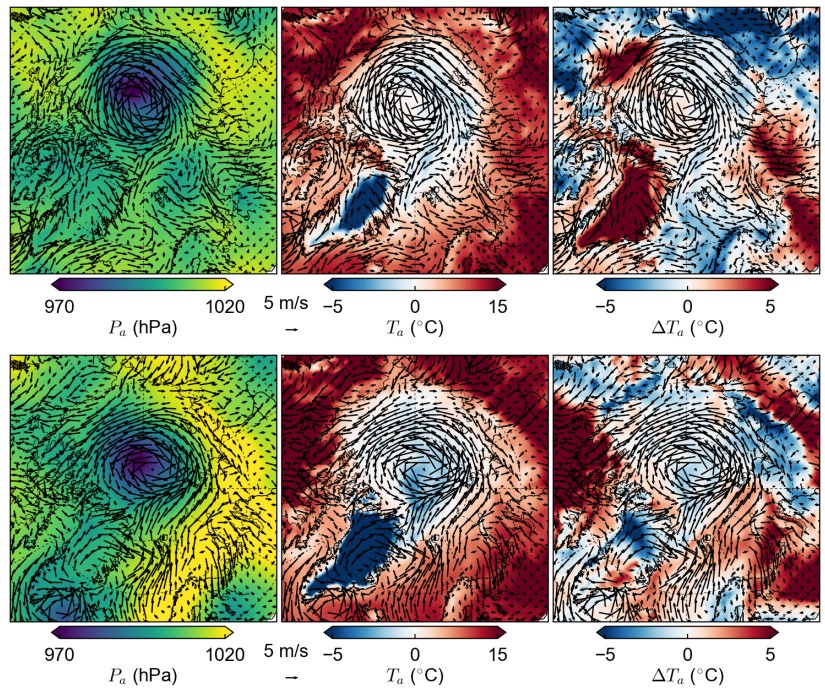

**Figure 11**: Sea level pressure (left), near-surface air temperature (middle) and near-surface air temperature anomaly relative
to the 2000-2016 mean (right) for the peak summer Arctic storm time periods in 2012 (top, Aug 6th) and 2016 (bottom, Aug
16th) from NASA's MERRA-2 reanalysis. The black vectors show the 10 m winds.





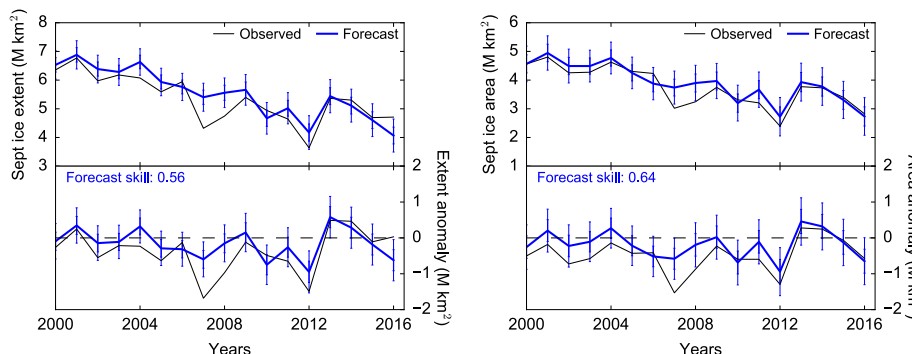

**Figure 12:** Forecasts of monthly sea ice extent (SIE, left) and sea ice area (SIA, right), generated using pan-Arctic NASA Team ice concentration data in June, following the spatial weighting forecast methodology described in Petty et al., (2017). The bottom panels show the anomalies relative to linear trend persistence. The skill value $S = 1 - (\sigma^2_{ferr}/\sigma^2_{anom})$, where

5  $\sigma_{ferr}$ ($\sigma_{ferr}$) is the root mean squared error of the linear trend persistence anomaly (forecast anomaly), calculated from 2000 to 2016. Note that the vertical lines indicate a one standard deviation confidence interval of the given forecast.

