# Peer review of "The Arctic sea ice cover of 2016: A year of record low highs and higher than expected lows"

_The Cryosphere, 2017_

## Referee Comment (RC1) · Anonymous Referee #1 · 15 Nov 2017

The Arctic sea ice cover displayed some unusual behavior in 2016. This submission explores a number of issues associated with this behavior. Part of the paper is devoted to a straight review of the year, while other parts delve into more scientific components. I think the mix works well. In a bigger picture it also present a detailed analysis and comparison between the main data sets usually used for Arctic ice studies. and how the choice of sets can, in some cases, affect the specific of conclusions. The authors present informed and careful analyses, and explain the associated uncertainties.

The has a strong potential to contribute substantially to the relevant literature. I suggest revision in accord with the points aired below.

Page 2, l 7: As part of this survey include recent analysis of Simmonds 2015 - Comparing and contrasting the behaviour of Arctic and Antarctic sea ice over the 35-year

period 1979-2013, Ann. Glaciol., 56(69), 18-28.

Page 4, line 17: NSIDC have recently released Version 3 of the sea ice data set analyses; see A. Windnagel, M. Brandt, F. Fetterer and W. Meier, 2017: Sea Ice Index Version 3 Analysis. NSIDC Special Report 19. National Snow and Ice Data Center, 80 pp. https://nsidc.org/sites/nsidc.org/files/files/NSIDC-special-report-19.pdf. The new version, and the reasons for it, are pertinent to some of the discussion (of differences) in the present investigation. Comments should be made (perhaps later) on this matter.

P 6, ll 21-23: Please to make a few brief words in connection with the use of the (new) MERRA-2 reanalysis, and how it compares with Version 1 of Rienecker, M. M., et al. (2011), MERRA: NASA's Modern-Era Retrospective Analysis for Research and Applications, J. Climate, 24, 3624–3648. Of relevance to the present investigation make mention, in particular, of the assimilation of satellite obs. not used in MERRA and the improvement in representations of the cryosphere.

Page 13, lines 27-28: We need some references here and quantification. Suggest referring to Montiel F, Squire VA (2017) Modelling wave-induced sea ice break-up in the marginal ice zone. Proceedings of the Royal Society A 473: 20170258 doi: 10.1098/rspa.2017.0258 and Kohout AL, Williams MJM, Toyota T, Lieser J, Hutchings J (2016) In situ observations of wave-induced sea ice breakup. Deep-Sea Research Part II, 131: 22-27 doi: 10.1016/j.dsr2.2015.06.010.

---

## Referee Comment (RC2) · Anonymous Referee #2 · 7 Dec 2017

Petty et al. revisited the unusual 2016 Arctic sea ice conditions by looking at the Arctic sea ice area (SIA) and extent (SIE), the compactness (SIA over SIE), the concentration budget (the ice intensification and ice divergence), sea surface temperature, and weather conditions. Comparisons were made with the 2000-2015 climatology. When calculating SIA and SIE, they evaluated the differences caused by different averaging methods and retrieval algorithms. They demonstrated that the choice of the averaging method could cause differences as large as the choice of retrieval algorithm. Although SIA and SIE differ with averaging method and retrieval algorithm, they show in common that the sea ice low anomalies at the start of 2016 did not translate low anomalies in summer. However, a record low of sea ice compactness was seen in summer 2016, which was likely caused by the two cyclones entering the Arctic Ocean in August. The

location and strength of the cyclones made them not able to melt out the sea ice and create a record low summer SIE.

The study has the potential to contribute to the understanding of the unusual behavior of Arctic sea ice in 2016. I suggest several revisions as follows.

1. L19, P1: Shouldn't the 'compactness' be sea ice area over sea ice extent, not only 'the estimates of sea ice area'?

2. L21, P2: The statement of 'a new record low September Arctic SIE was not suggested by the SIO in 2016, despite this strong winter/spring preconditioning' seems not objective enough without mentioning whether these models in SIO could successfully predict the winter/spring preconditioning as strong as observed.

3. L30, P3: The study only used Bootstrap SIC data for the year 2016. Is it because the NASA Team data was not available? Please clarify.

4. Section 2.2 lacks details of how the ice drift data will be used in the following study.

5. L1, P8: From the Figure 3, it is very difficult to see the differences in SIE between the NASA Team and Bootstrap data. The differences should refer to Table 1 instead.

6. L23, P8: This paragraph seems subjective. Any literature review on quantifying the differences between the two products from the perspectives mentioned in this paragraph?

7. L3, P8: Suggest replace 'methodology' with 'averaging methodology', and 'algorithm' with 'retrieval algorithm' for readability. This could apply to the whole paragraph.

8. Suggest the authors be more careful with delivering the results. For example, in Line 13, Page 10, negative anomalies in the Bering seas are seen in Jan, not obvious in Feb and Mar. And positive anomalies in the Labrador Sea and the Sea of Okhotsk are not clear with the black sea ice edge lines. Another example is in line 11, Page 11, strong positive anomalies are seen in the Chukchi Sea in both November and December,

which is not consistent with the statement of 'the autumn SIC anomalies are mainly negative'.

- 9. Line 26, Page 10: a similar pattern to what pattern?
- 10. Section 4.4: All the referred Figure 10 should be Figure 11.
- 11. L24, P14: This sentence is confusing.

СЗ

---

## Author Comment (AC1) · 21 Dec 2017

**Author response to interactive comments on: "The Arctic sea ice cover of 2016: A year of record low highs and higher than expected lows" by A. A. Petty et al.**

Reviewer comments are in black, our responses are in blue.

We include a pdf of the new manuscript and a word document highlighting the tracked changes we have made based on these comments.

**Anonymous Referee #1

The Arctic sea ice cover displayed some unusual behavior in 2016. This submission explores a number of issues associated with this behavior. Part of the paper is devoted to a straight review of the year, while other parts delve into more scientific components. I think the mix works well. In a bigger picture it also present a detailed analysis and comparison between the main data sets usually used for Arctic ice studies. and how the choice of sets can, in some cases, affect the specific of conclusions. The authors present informed and careful analyses, and explain the associated uncertainties.

The has a strong potential to contribute substantially to the relevant literature. I suggest revision in accord with the points aired below.

We sincerely thank Reviewer #1 for taking the time to review the manuscript and provide these comments. See below for the comments and our responses.

Page 2, l 7: As part of this survey include recent analysis of Simmonds 2015 - Comparing and contrasting the behaviour of Arctic and Antarctic sea ice over the 35-year period 1979-2013, Ann. Glaciol., 56(69), 18-28.

The above paper is a useful assessment of Arctic/Antarctic sea ice cover over recent decades, but we're unsure what from that paper we are being asked to cite and where this should go in the manuscript (L1, 7 or 17 don't make sense!). Happy to do this if the reviewers could provide further clarification.

Page 4, line 17: NSIDC have recently released Version 3 of the sea ice data set analyses; see A. Windnagel, M. Brandt, F. Fetterer and W. Meier, 2017: Sea Ice Index Version 3 Analysis. NSIDC Special Report 19. National Snow and Ice Data Center, 80 pp. https://nsidc.org/sites/nsidc.org/files/files/NSIDC-special-report-19.pdf. The new version, and the reasons for it, are pertinent to some of the discussion (of differences) in the present investigation. Comments should be made (perhaps later) on this matter.

Yes definitely, we were expecting to do this (Walt Meier is a co-author of this paper and the new v3.0 dataset).

At P4, L17 we have added: ' Note that a new, version 3.0, Sea Ice Index (Fetterer et al., 2017) was released by the NSIDC during the discussion phase of this study, as discussed below.'

At P5, L10 we have adapted the discussion to read: 'During the discussion phase of this paper the NSIDC, as expected, switched to using this new methodology for their new, version 3.0, Sea Ice Index (Fetterer et al., 2017), making a comparison of these different approaches timely. A detailed assessment of the differences between the version 2.0 and 3.0 indices are provided in the accompanying NSIDC special report (Windnagel et al., 2017).'

At P9, L2 we have added: '(equivalent to the new version 3.0 NSIDC Sea Ice Index)'

P 6, ll 21-23: Please to make a few brief words in connection with the use of the (new) MERRA-2 reanalysis, and how it compares with Version 1 of Rienecker, M. M., et al. (2011), MERRA: NASA's Modern-Era Retrospective Analysis for Research and Applications, J. Climate, 24, 3624–3648. Of relevance to the present investigation make mention, in particular, of the assimilation of satellite obs. not used in MERRA and the improvement in representations of the cryosphere.

Agreed. We have added the following to the revised manuscript: 'MERRA-2 offers several improvements over the original MERRA reanalysis, including: the assimilation of additional satellite observations (e.g. space-based observations of aerosols, modern hyperspectral radiance and microwave observations), the use of daily sea ice and SST fields (compared to weekly fields in MERRA) and a seasonally varying (instead of a constant) surface albedo (Cullather and Bosilovich, 2017)'.

Page 13, lines 27-28: We need some references here and quantification. Suggest referring to Montiel F, Squire VA (2017) Modelling wave-induced sea ice break-up in the marginal ice zone. Proceedings of the Royal Society A 473: 20170258 doi: 10.1098/rspa.2017.0258 and Kohout AL, Williams MJM, Toyota T, Lieser J, Hutchings J (2016) In situ observations of wave-induced sea ice breakup. Deep-Sea Research Part II, 131: 22-27 doi: 10.1016/j.dsr2.2015.06.010.

We have added the following references we believe summarize recent evidence regarding storm/wave-induced sea ice break-up, ocean mixing, and sea ice loss: '(e.g. Zhang et al., 2013, Kohout et al., 2014, Kohout et al., 2016, Montiel and Squire 2017)'. We believe an attempt to quantify the direct impact of this storm on waves-ocean mixing-sea ice loss to be beyond the scope of this paper (a study in and of itself!).

**Anonymous Referee #2

Petty et al. revisited the unusual 2016 Arctic sea ice conditions by looking at the Arctic sea ice area (SIA) and extent (SIE), the compactness (SIA over SIE), the concentration budget (the ice intensification and ice divergence), sea surface temperature, and weather conditions. Comparisons were made with the 2000-2015 climatology. When calculating SIA and SIE, they evaluated the differences caused by different averaging methods and retrieval algorithms. They demonstrated that the choice of the averaging method could cause differences as large as the choice of retrieval algorithm. Although SIA and SIE differ with averaging method and retrieval algorithm, they show in common that the sea ice low anomalies at the start of 2016 did not translate low anomalies in summer. However, a record low of sea ice compactness was seen in summer 2016, which was likely caused by the two cyclones entering the Arctic Ocean in August. The location and strength of the cyclones made them not able to melt out the sea ice and create a record low summer SIE.

The study has the potential to contribute to the understanding of the unusual behavior of Arctic sea ice in 2016. I suggest several revisions as follows.

We sincerely thank Reviewer #2 for taking the time to review this manuscript and provide the following comments. See below for our responses.

1. L19, P1: Shouldn't the 'compactness' be sea ice area over sea ice extent, not only 'the estimates of sea ice area'?

Agreed, we have changed this to sea ice area over sea ice extent

2. L21, P2: The statement of 'a new record low September Arctic SIE was not suggested by the SIO in 2016, despite this strong winter/spring preconditioning' seems not objective enough without mentioning whether these models in SIO could successfully predict the winter/spring preconditioning as strong as observed.

We think the reviewer is asking us how well the models 'captured' or 'simulated' the winter/spring conditions (rather than predicted) and how that relates to forecasting the summer sea ice extent? This is a challenge, as the SIO doesn't necessarily provide information regarding the springtime conditions used to drive the individual forecasts. Indeed for the dynamical models, carrying out a thorough assessment of how well they are capturing the observed winter/spring conditions before their forecasts are generated would be a lot of work (and another study in and of itself).

We have attempted to make it clearer that we are referring to the observed strong winter/spring preconditioning by adding 'seen in the observations' at the end of this sentence to make clear we are not saying we believe they are necessarily capturing the winter/spring conditions.

3. L30, P3: The study only used Bootstrap SIC data for the year 2016. Is it because the NASA Team data was not available? Please clarify.

We do use 2016 NASA Team data (NRT), although we focus more on the Bootstrap data/analysis as explained later in the manuscript. We have updated that line to read: 'Note that for 2016 we use the daily near real-time (NRT) NASA Team SIC data (Maslanik, J. and J. Stroeve. 1999) and daily Bootstrap SIC data (provided courtesy of J. Comiso).'

4. Section 2.2 lacks details of how the ice drift data will be used in the following study.

How the ice drifts are used in the concentration budgets is discussed in the following methodology section. We have changed '(methodology discussed in the following section)' to '(methodology discussed in Section 3)' to make this clearer.

We did also add to Section 2.2 the following: ' The mean 2016 monthly KIMURA ice drifts are shown in Figure 1, which are produced by averaging the daily ice drifts within each month.'

5. L1, P8: From the Figure 3, it is very difficult to see the differences in SIE between the NASA Team and Bootstrap data. The differences should refer to Table 1 instead.

The 2016 values are also included in Figure 3 though, as in Table 1, so we think this is clear from the figure alone. We added the table for those more interested in seeing just the numbers presented alongside each other. We hope this is satisfactory.

6. L23, P8: This paragraph seems subjective. Any literature review on quantifying the differences between the two products from the perspectives mentioned in this paragraph?

As stated in the data section (2.1), the differences between the two products have been well explored in previous studies. Here we wanted to just list a few of the pertinent issues regarding this data the reader may be interested in knowing for this study. We have added the following to the end of this paragraph: ' As stated earlier (Section 2.1), the differences between the NASA Team and Bootstrap data have been well documented (e.g., Comiso et al., 1997; Meier, 2005; Ivanova et al., 2015; Comiso et al., 2017) and we refer the reader to these studies for more information regarding the differences between the two algorithms and data products.'

7. L3, P8: Suggest replace 'methodology' with 'averaging methodology', and 'algorithm' with 'retrieval algorithm' for readability. This could apply to the whole paragraph.

Agreed, we have replaced 'methodology' with 'averaging methodology' throughout the paper and added 'retrieval algorithm' where appropriate to improve readability.

8. Suggest the authors be more careful with delivering the results. For example, in Line 13, Page 10, negative anomalies in the Bering seas are seen in Jan, not obvious in Feb and Mar. And positive anomalies

in the Labrador Sea and the Sea of Okhotsk are not clear with the black sea ice edge lines. Another example is in line 11, Page 11, strong positive anomalies are seen in the Chukchi Sea in both November and December, which is not consistent with the statement of 'the autumn SIC anomalies are mainly negative'.

We think some of this confusion might have come from the fact we are discussing both SIC anomalies (Figure 2) and intensification anomalies (Figure 6). We have attempted to make our results clearer by first adding in some notes regarding the figures being discussed (e.g. Figure 2), then by adding some changes/more detail as recommended, e.g. we added ' the Sea of Okhotsk (in February and March)'. We do feel that the Bering Sea SIC anomalies can be seen in all 3 winter months, however, so we kept that as is.

9. Line 26, Page 10: a similar pattern to what pattern?

We have changed this to: ' The April results show similar spatial patterns of SIC anomalies to winter'

10. Section 4.4: All the referred Figure 10 should be Figure 11. 11.

We have updated these to Figure 11.

L24, P14: This sentence is confusing.

We have changed this to: 'Similar to 2012, the median July SIO forecast of September SIE was biased high.'